# Effect of Behavioral Precaution on Braking Operation of Elderly Drivers under Cognitive Workloads

**DOI:** 10.3390/s22155741

**Published:** 2022-07-31

**Authors:** Yusuke Kajiwara, Eri Murata

**Affiliations:** Department of Production Systems Engineering and Sciences, Komatsu University, Shichomachi Nu1-3, Komatsu 923-8511, Japan; 18111066@komatsu-u.ac.jp

**Keywords:** braking mistakes, elderly driver, cognitive stress, action slip, coping skills

## Abstract

The number of accidents by elderly drivers caused by the erroneous tread of a brake pedal or accelerator pedal has increased. A recent study reported that the number of accidents could be reduced by preparing for braking mistakes due to driving behavior by using a simulator. However, related studies have pointed out that driving behavior in simulators does not always reflect driving behavior in the real world. This paper focuses on the posture of the left foot as a behavioral precaution and provides insights into braking mistakes by comparing behavioral precautions taken on simulators and on public roads. In the experimental results, cognitive and action errors increased with age, but elderly drivers are less likely to have an accident when they are exposed to the risk of collision in situations with a mental workload by making space for the right foot to step on the brake pedal. Elderly drivers with coping skills had their left foot perpendicular to the ground and their body was unstable. This result was different from the driving behavior in the simulator, but it was not possible to identify that this difference was the cause of the collision accidents. Coping skills were predicted with 70% accuracy from the left foot posture of an elderly driver near the intersection. We expanded the system’s range of use and enhanced its usefulness by predicting coping skills derived from natural driving behavior in the real world. The contributions of this study are as follows. We clarify the effect of behavioral precautions on the braking operation of elderly drivers when under a cognitive workload. We provide new insights into the use of behavioral precautions in older drivers’ braking operations in the real world. We predicted coping skills from natural driving behavior near intersections in the real world.

## 1. Introduction

The number of accidents involving elderly drivers caused by an erroneous tread on the brake pedal or accelerator pedal has increased [1]. Therefore, it is necessary to prevent mistakes in braking operations by elderly drivers. Elderly drivers show a decline in cognitive ability and physical functions such as working memory [2], visual attention [3], visual processing speed [4], ability to adapt to new sensory information [5], muscle action [6], and reaction times [7]. These factors increase elderly drivers’ exposure to the risk of collision in situations with a mental workload. A related work [8] analyzed the driving behavior of an elderly driver using a driving simulator and suggested that elderly drivers who felt impatience and nervousness could not properly step on the brake pedal. In addition, elderly drivers who took behavioral precautions, such as stabilizing their bodies with their left foot in case of an unexpected situation, had fewer accidents in the simulator [8]. The ability to prepare for this unexpected situation in advance was defined as a coping skill. However, a related study measured and analyzed only driving behavior in the simulator. Driving behavior on the simulator does not always match driving behavior in the real world. In addition, the effect of cognitive decline in elderly drivers regarding braking and behavior precautions under mental stress was not analyzed [9].

In this paper, driving behavior in the simulator is compared with driving behavior at intersections on the public roads, and the relationship between driving behavior on public roads and coping skills is investigated. We expand the scope of the system and enhance its usefulness by predicting coping skills by looking at elderly drivers’ prepa-rations for braking operations near intersections.

Experimental results show that elderly drivers with many cognitive and action errors did not necessarily have many accidents. The reason for this is that elderly drivers were aware of the decline in their cognitive and physical function and adapted their driving behaviors in response. Typical examples of adaptive driving behaviors were driving at slow speeds and adjusting the posture of the left foot to more easily step on the brake pedal with the right foot. However, it should be noted that low-speed driving is a risk-taking behavior and increases the risk of collision in some situations [10,11]. There was a medium correlation between the posture of the left foot when driving on the simulator and the posture of the left foot when driving on public roads. In addition, coping skills were predicted with 70% accuracy from the posture of the elderly driver’s left foot near the intersection.

## 2. Related Work

### 2.1. Cognitive Precautions and Behavioral Precautions

If a driver’s attention is restricted, this increases the risk of collisions [12]. Elderly drivers pay less attention due to aging and tend to miss traffic signs [13]. The reaction time in driving operations is also slowed by the use of mobile phones and distractions in the interior of the vehicle [10,11,12,13,14,15,16,17]. These factors increase the risk of driving errors and collisions for elderly drivers [13,18]. These related studies suggest that distractions to older drivers cause them to miss traffic signs and increase the risk of collisions. Therefore, it is important for older drivers to pay attention to the surrounding road conditions in order to reduce the risk of collision. Paying attention to surroundings is a cognitive precaution that the driver can take to avoid a collision. However, elderly drivers not only show a decline in cognitive function but also have weakened muscle strength [6]. Therefore, elderly drivers should take behavioral precautions in addition to cognitive precautions to prevent braking mistakes. Behavioral precautions are advance preparations such as stabilizing the body and making space so that the brakes can be accurately and easily applied. A related study [8] defined coping skills as the ability to take appropriate behavioral precautions. Coping skills are different from a working memory and attention. Elderly drivers with coping skills can be aware of their cognitive and physical decline and prepare in advance to respond to the unexpected situation. The group with coping skills was defined as CS, while an elderly driver with a poor driving performance under a mental workload is defined as NCS. A Venn diagram of the CS and NCS groups is shown in Figure 1. Elderly drivers in the CS group stabilized their bodies with their left foot. Coping skills were predicted with 92% accuracy using a machine learning method that learned the characteristic elements of driving behavior in the simulator. However, the characteristic driving behaviors in the simulator that were revealed in the related study may not represent driving behavior in the real world. A related study [9] reported that cognitive load adversely affected braking operations. The effect of taking behavioral precautions on braking when a cognitive load is applied has not been verified.

### 2.2. Contribution of This Study

The contributions of this study are as follows.

We clarify the effect of behavioral precautions on the braking operations of elderly drivers when under a cognitive workload.We provide new insights into the use of behavioral precautions in older drivers’ braking operations in the real world.We predicted coping skills from natural driving behavior near intersections in the real world.

A related study [9] reported that the drivers’ braking operations slow down when a driver has a cognitive workload. However, this has not been validated for behavioral precautions when braking is impaired by a cognitive workload. This study analyzes the dual-task performance of older drivers and reveals the effects of behavioral prevention on drivers braking under cognitive workloads.

It is difficult to perfectly reproduce a real-world situation on the simulator. For example, when the driver accelerates the car in the real world, the driver feels the acceleration and stabilizes the body with his left foot. In addition, the position and posture of the left foot of the elderly driver will differ depending on the vehicle type. This study compares driving behavior on a simulator with driving behavior in the real world. It provides insights into real-world behavioral precautions that have not been verified in related studies [8].

Elderly drivers are most likely to mistake the brake and accelerator when starting from a stopped state [1]. When a driver drives near an intersection, he/she feels cognitive stress, impatience, and tension. For this reason, we focused on preparing for braking near intersections. In a related study [8], coping skills could not be predicted without driving in a special environment, such as a simulator. We expanded the range of use of the coping skill prediction system and enhanced its usefulness by making it possible to predict the natural driving behavior near an intersection.

## 3. Expansion of Coping Skills Prediction System

From Figure 2, we expand the coping skill prediction system of the related work [8] and predict coping skills based on the preparations made for a braking operation at intersections on public roads. The coping skill prediction system of related works predicts coping skills from the preparations made for a braking operation when an elderly driver is nervously driving on the simulator. We focused on the left foot posture as preparation for braking operation [8]. The training data were the tilt angle and angular velocity of the left foot of the driver, who drove the route on the simulator within the time limit. The driver felt impatient and nervous because the simulator must be completed within the time limit. However, there are few cases where a driver hastily drives from a departure point to a destination on a public road within a time limit.

In this study, we focused on the preparation for the braking operation when the elderly drivers drove near an intersection on the public road. The driver must complete driving operations such as turning left or right before the traffic light changes from green to red. At that time, the driver pays attention to moving obstacles such as pedestrians and oncoming vehicles. As a result, when elderly drivers drive near an intersection on the public road, they are cognitively stressed and feel impatient and nervous. Therefore, the coping skills’ prediction system inputs the left foot posture near the intersection into machine learning to predict coping skills. In this study, we used the random forest and naïve Bayes methods to evaluate the accuracy of coping skill predictions.

## 4. Experimental Design

### 4.1. Purpose

We conducted three types of experiments: driver’s reaction to cognitive stress, driving on a simulator and driving on a public road. In this study, we used the dual task to analyze the effect of cognitive workload on the braking operation of elderly drivers. In addition, we analyzed the dual-task performance of elderly drivers regarding the number of accidents that occurred in the simulator. We analyzed the effect that elderly drivers preparing for the braking operation had on the number of accidents in the simulator. We expanded the scope of the coping skills prediction system and enhanced its usefulness by classifying CS and NCS from elderly drivers’ preparation for the braking operation near intersections. After obtaining informed consent, the experiment was conducted. The experiment was conducted between October 2021 and December 2021. The subjects of this study were completely different from the subjects of the related study [8]. Regarding the experimental design, the acquisition of driving behavior in the simulator was performed using the same procedure as in the related research [8]. The experimental design regarding driving behavior in dual tasks and driving behavior in the real world was added and carried out in this study only.

### 4.2. Subjects

The subjects included 50 drivers. The age group consisted of 32 drivers aged 20–64 years (mean ± SD = 32.4 ± 16.8) and 18 drivers aged 65–79 years (mean ± SD = 70.9 ± 4.4). Drivers from 20 to 64 years were defined as the adult driver group. Drivers from 65 to 79 ages were defined as the elderly driver group. A cognitive ability test was performed on subjects in the elderly driver group. Following a cognitive ability test, the elderly drivers were found to have no problem with cognitive function.

### 4.3. Equipment

Table 1 shows the specifications of the equipment used in the experiment and the measurements. The mean, standard deviation, maximum, and minimum values of the variables measured in Table 1 are analyzed. IMU measures the movement of the subject’s head and left foot during the experiment. Elderly drivers wore IMUs on their ankles. The knee was in the negative direction of the IMU’s *y*-axis. The tilt angle on the *x*-axis represents the tilt angle of the left foot. A tilt angle of −90 degrees on the *x*-axis means that the tilt angle of the left foot is perpendicular to the ground. A positive tilt angle on the *y*-axis means that the drivers turn their toes inwards.

We analyzed the driver’s 2D gaze position when object detection detected a moving obstacle and a traffic sign. There are three types of moving obstacles: vehicles, pedestrians, and bicycles. There are five types of traffic signs: traffic lights, pedestrian crossing signs, stop signs, speed limit signs, and railroad crossings. The eye tracker measured the eye-gaze of a driver while driving on a public road. An eye tracker was worn on the head of a driver. The pupil core was equipped with two eye cameras and one front camera. The eye cameras captured an image of both eyes. The front camera captured the image at which the driver was looking. To calculate the 2D gaze position, the pupil analysis software “pupil capture” was used. Pupil capture calculated the 2D gaze position from eye images and views images. The accuracy of the 2D gaze position was 0.60°. The precision of the 2D gaze position was 0.02. The 2D eye-gaze coordinates are represented in the normalized image coordinate system. The center of the view image was 0 on the XY axes of the normalized image coordinate system. The *X*-axis represents the horizontal eye-gaze movement. The *Y*-axis represents vertical eye-gaze movement. The 2D eye-gaze position was represented in the range from −1 to 1 on the XY axes. We detected the objects in the view image using machine learning. The object detection algorithm was yolov5. The yolov5 program was downloaded from git hub (https://github.com/ultralytics/yolov5 (accessed on 6 February 2022)). The program was executed using the Visual studio code. The Python version is 3.7. The computer was equipped with GPU: RTX3080. We trained the yolov5s model on a custom dataset. Custom datasets comprised annotation data created from view images. The number of custom datasets was 5800. We detected 10 types of objects: side-view mirror, rear-view mirror, speed meter, vehicle, traffic light, crosswalk sign, stop sign, speed limit sign, railroad crossing, pedestrian, and bicycle. Figure 3, Figure 4 and Figure 5 show the object detection accuracy. The object detection precision was mAP@0.5 with 0.893. The object detection precision and the 2D gaze position precision were sufficient to analyze eye-gaze near moving obstacles and traffic signs.

The vehicle speed was calculated from the position of the vehicle. The position of the vehicle was acquired using the GNSS sensor mounted on the smartphone. The position of the vehicle is represented by the longitude and latitude. The vehicle speed is calculated from the latitude and longitude (*x*_1_, *y*_1_) of point A at time *t*_1_ and the latitude and longitude (*x*_2_, *y*_2_) of point B at time *t*_2_, as follows:(1)$$d=\frac{r\cos^{-1}(\sin y_{1}\sin y_{2}+\cos y_{1}\cos y_{2}\cos(x_{2}-x_{1})}{t_{2}-t_{1}},$$
where *r* is the radius of the earth, which is 6378.137 km. To reduce noise, a moving average filter was applied to the time series data calculated by Equation (1). The time window lasted 3 s.

### 4.4. Dual Task

We used the dual task to verify the driver’s reaction under cognitive stress. The measurement environment for the dual task is shown in Figure 6. The experiment was conducted indoors. The subject wore an IMU sensor on his/her left foot.

The subject solves the calculation task and, at the same time, performs the driving behavior according to the visual stimulus shown on the display. Calculation tasks are processed in the working memory [19] and long-term memory. The working memory consists of a central executive and a slave system. The central execution controls attention, integrates information, and manages slave systems. The slave system temporarily retains and manipulates information. The slave system consists of a phonological loop, visuospatial sketchpad, and episode buffer. As the driver ages, the associative memory, memory retention/retrieval, attention suppression function, and attention target updates decline [20,21,22]. The performance of a dual task represents an overall evaluation of these functions. Driving behavior has two patterns: “stepping from the accelerator to the brake” (pattern A) and “turning the steering wheel to the right” (pattern B). The initial state of Pattern A is the state of stepping on the accelerator. The initial state of Pattern B is the natural state of the steering wheel. There are two types of behavioral task: a simple task that is performed only in pattern A and a complex task that combines pattern A and pattern B. The subject is cognitively burdened when solving a calculation task. Subjects calculated the addition of two or three digits. The subject listened to the formula played by the voice recorder and calculated the sum in his head according to the formula. If the subject was able to calculate in time, the subject verbally responded with the calculation result. If the subject was unable to calculate in time, he/she was instructed to preferentially solve the next calculation task. The formulas were played at intervals of once every 7 s. Figure 7 shows the stimulus and the stimulus interval. The stimuli are the red and green circles on the display. The initial screen is blank. The circle is displayed for 3 s. When the subject recognizes the red circle, the subject acts as pattern A. When the subject recognizes the green circle, the subject acts as pattern B. As soon as the subject recognized that the circle had disappeared, the subject was instructed to return to the initial state of action. The circles were set to appear on the display at intervals of 11–14 s. The circle was displayed seven times in the simple and complex tasks. In the complex task, green or red circles were randomly presented.

We defined the simple task and complex task errors as follows:(a)The subject did not act when the circle was displayed. The subject acted after the circle was displayed.(b)The subject did not return to the initial state for more than 3 s after the action.(c)Subjects acted improperly in response to the presented stimuli.(d)The number of errors was counted by visually checking the action.

We analyzed the reaction under cognitive stress by dividing reactions into cognitive errors and action errors. Cognitive errors were defined as mistakes caused by not noticing the stimulus or delays in noticing the stimulus. Action errors mean action slips [23] and are defined as mistakes caused by incorrect schema activation or incorrect triggers. From the definition of cognitive error, the number of cognitive errors is the sum of the number of errors of types (a) and (b). From the definition of action error, the number of action errors is the sum of the number of errors of types (c) and (d).

### 4.5. Driving on the Simulator

The experimental design for driving on the simulator was the same as in the related study [8]. The measurement environment for driving on the simulator is shown in Figure 8. The subject wore an IMU sensor on his/her left foot.

Elderly drivers were classified into CS and NCS groups based on the number of accidents while driving in a driving simulator, using the same thresholds as in the related study [8]. We instructed the subjects to drive the route within 10 min. Drivers felt impatient and nervous because the route must be completed within the time limit. The driver drove on the route containing the scenes for the first time. The 15 cases in Table 2 show unexpected situations, since the driver was driving on an unfamiliar road. Therefore, by conducting the experiment, the subjects’ coping skills can be measured. Subjects with a small number of accidents in a scene with a mental workload have high coping skills. An accident is a collision with a pedestrian or a vehicle. The steering operation was measured from −720 degrees to 720 degrees. In addition, the brake and accelerator represent the depressed state, which can range from 0% to 100%.

### 4.6. Driving on Public Roads

The driver drove to two locations: route A and route B. The driving route is shown in Figure 9. Double circles in the figure indicate intersections. Figure 10 shows the measurement environment when drivers drove on public roads. The subject wore an IMU sensor on his/her left foot and head. The subject wore an eye-tracker on his/her head. A smartphone with a GNSS sensor was in the vehicle. The tilt angle of the *z*-axis of the head indicates left–right confirmation. When the vehicle stopped, we excluded it from the analysis data because this contains many secondary tasks that are unnecessary for driving. However, since the value measured by GPS contains noise, “vehicle stop” in this study means that the vehicle speed was 5 km/h or less.

### 4.7. Analysis

In this study, we use a hypothesis test to analyze driver characteristics using the Mann–Whitney U test. Many studies test both the normally distributed population and equal SDs assumptions through statistical significance tests [24]. However, with a small sample size, the power may be insufficient to detect a significant difference between the sample data and the normal distribution. In this study, we adopt a nonparametric test to avoid the problem. The Mann–Whitney U test is non-parametric. The Mann–Whitney U Test has high statistical power, even in small samples. The variables were the mean, standard deviation, maximum value, and minimum value of the measured values. The null hypothesis was “there is no difference in the characteristic elements of both groups”. The rejection region was 0.01. When the null hypothesis is rejected, a variable shows a significant difference between the two groups. However, it has been pointed out that the significance level is not an absolute indicator [25]. Therefore, the characteristic elements with a *p*-value less than 0.05 were also analyzed. The subjects were divided into two groups according to age, number of errors, and number of accidents. The subjects were divided into the adult driver group and the elderly driver group. The adult driver group consisted of drivers under the age of 64. The elderly driver group consisted of drivers over the age of 65. Subjects in the elderly driver group were divided into groups that made a few errors and groups that made many errors. We set the threshold for the sum of cognitive errors to 2, we set the threshold for the sum of action errors to 2, and we set the threshold for the sum of errors to 5. Subjects in the elderly driver group were divided into the NCS group and CS group. In the related study, the average number of accidents on the simulator was 1.6 ± 1.1, so subjects with two accidents or fewer were categorized as CS and other subjects were categorized as NCS. In this experiment, CS and NCS were classified using the same threshold value (=2) as in the related studies.

## 5. Results

### 5.1. Detection Response Task

Figure 11 shows the number of dual tasks errors made by each age group. As the subject ages, the number of errors in the simple task and complex tasks increases. The adult drivers made no cognitive errors, but the elderly drivers made cognitive errors. Drivers over the age of 40 had more than double the number of action errors compared to drivers under the age of 40. The number of cognitive errors in a simple task and a complex task was the same. The subjects made a higher number of action errors in the complex task than in the simple task. We defined the sum of cognitive errors as the sum of cognitive errors in a simple task and a complex task, we defined the sum of action errors as the sum of action errors in the simple task and the complex task, and we defined the sum of errors as the sum of cognitive errors and action errors in the simple task and the complex task. We analyzed the difference in the sum of errors between the adult driver group and the elderly driver group using the U test. The test results show that the sum of errors was larger for elderly drivers than adult drivers. We analyzed the difference in the left foot posture between the low-errors group and the many-errors group using the U test. The subjects were categorized into the low-errors group and the many-errors group based on the sum of the errors. The results of the U test are shown in Table 3. The correlation coefficient between the sum of cognitive errors and the sum of action errors was 0.57, showing a medium correlation.

### 5.2. Driving on the Simulator

We analyzed the difference in the number of dual-tasks errors between the NCS group and CS group using the U test. Table 4 shows the number of dual-tasks errors for the CS and NCS group. We analyzed the difference in driving operations between the low-errors group and the many-errors group using the U test. Table 5 shows the driving operation for CS and NCS groups.

We analyzed the difference in driving operation between the adult driver group and the elderly driver group using the U test. Table 6 shows the driving operation for adult drivers and elderly driver groups.

### 5.3. Driving on the Simulator

We analyzed the difference in vehicle speed between the adult driver group and the elderly driver group using the U test. Table 7 shows the vehicle speed for adult drivers and elderly driver groups. The difference between the two groups in terms of the standard deviation of the vehicle speed was statistically significant. The *p*-value of the mean of the vehicle speed was less than or equal to 0.05. Therefore, the difference between the two groups in terms of the mean vehicle speed was probably statistically significant.

We analyzed the difference in the head movements and eye-gaze between the NCS group and the CS group using the U test. Table 8 shows the head movements and the eye-gaze for NCS and CS groups.

We analyzed the difference in the left foot posture between the NCS group and the CS group using the U test. Table 9 shows the left foot posture for NCS and CS groups.

The correlation coefficient of the average tilt angle of the left foot on the *y*-axis when driving in the simulator and driving in the real world showed a moderate correlation (=0.44). The correlation coefficient of the average tilt angle of the left foot on the *x*-axis when driving in the simulator and driving in the real world showed a moderate correlation (=0.51). The correlation coefficient of the average tilt angle of the left foot on the left *y*-axis when driving in the simulator and the dual task showed a strong correlation (=0.84). The correlation coefficient of the average tilt angle of the left foot on the *x*-axis when driving in the simulator and the dual task showed a strong correlation (=0.99).

The characteristics of the left foot posture in Table 9 were input for machine learning. The elderly drivers were classified into NCS and CS. The classification results are shown in Table 10 (F1 = 0.7).

## 6. Discussion

In Figure 6, elderly drivers are shown to have more cognitive errors than adult drivers. This result is consistent with reports from related work [13] showing that elderly drivers are more likely to miss traffic signs. The number of action errors doubles at ages 40 and older compared to 20–40 years. Elderly drivers made more action errors than cognitive errors. From the results, it could be suggested that action errors form the main factor that causes mistakes in braking operations. From the statistical data [1], the number of accidents in which the braking operation was carried out improperly increases by about 10 times in elderly drivers aged 75 and over compared to elderly drivers aged 74 and under. Salmon et al. [26] reported that improper driving operations occur due to observational failures and cognitive delays. From the results of the experiment and these findings, it is suggested that cognitive errors induce action errors. The sum of the errors represents a braking operation mistake caused by a combination of action errors and cognitive errors. From Table 3, elderly drivers with a small sum of errors were shown turn their left foot toes outward in a simple task. This position creates space for the right foot to step on the brake pedal. Therefore, elderly drivers can easily step on the brakes with their right foot. From Table 5, the elderly drivers with a large sum of action errors did not perform enough steps on the brake pedal. These results show that elderly drivers who took behavioral precautions were able to brake and operate properly, even under high levels of cognitive stress.

From Table 4, the difference between the CS and NCS group in terms of the number of dual-tasks errors was not statistically significant. From Table 5, drivers with a large sum of errors were driving unnecessarily slowly near an intersection where a child was likely to rush out into the street. The driver with a large sum of errors turned the steering wheel unnecessarily to avoid parked vehicles. In Table 6, the elderly driver group was driving at a slower speed than the adult driver group. These results show that elderly drivers adjust their driving behavior in response to cognitive decline. Therefore, there was no significant difference in the number of accidents between the many-errors group and the few-errors group because the elderly driver adjusted his/her driving behavior as a precaution. However, it should be noted that excessive speed limits and avoidance can increase the risk of collision, depending on the road conditions [10,11].

In Table 7, the adult driver can be seen to drive faster than the elderly driver. This result is consistent with the result of the simulator in Table 6. Therefore, it is suggested that elderly drivers adjusted their vehicle speed in response to cognitive decline when driving on a public road. From Table 8, when the elderly driver drove at the intersection on the public road, CS checked left and right more thoroughly than NCS. Left–right confirmation is a type of perceptual precaution. Jha, S et al. [27] reported that eye movements and head movements were correlated. Brown et al. [28] reported that left–right confirmation reduced the risk of collision. The results in Table 8 are consistent with the results of these related works.

We consider the predictability of coping skills based on driving behavior at intersections on public roads. From Table 10, the correlation coefficient of the left foot posture when driving on the simulator and the driving on the public road showed a medium correlation. In addition, the correlation coefficient of the left foot posture when driving on the simulator and the dual tasks showed a strong correlation. The reason why the posture of the left foot when driving in the simulator and driving in the real world showed a medium correlation is considered to be the difference in the environment inside the vehicle. The posture of the left foot is calculated from the accelerometer, angular velocity, and geomagnetic sensor. Therefore, it is possible that the acceleration of the car was mixed with the value acquired by the accelerometer, causing an error in the calculation of the posture of the left foot, but this effect was very small because the vehicle was driven at low speed near the intersection. From Table 9, CS in elderly drivers turned their left foot toes outward. To adapt to cognitive decline, the elderly drivers with a small sum of errors when performing dual tasks turned their left foot toes outward to create space for the right foot to step on the brake pedal. It can be expected that elderly drivers are less likely to have an accident on the public road when exposed to the risk of collision in situations with a mental workload if they take behavioral precautions. In addition, the mean tilt angle of the *X*-axis of the left foot is more perpendicular to the ground in CS than in NCS. On the other hand, the SD of the *X*-*axis* tilt angle of the left foot is larger in CS than in NCS, and the body is unstable. However, it was not possible to clarify whether this difference in the inclination angle of the left foot directly causes collisions in the real world. These results suggest that coping skills can be predicted from the left foot posture when elderly drivers drive at an intersection on a public road. The elderly drivers are classified into NCS and CS (F1 = 0.7). The classification accuracy of this study is inferior to that of a related study [8]. However, since the coping skills can easily be predicted from driving behavior on public roads, the coping skills prediction system in this study is more useful than that in the related study [8]. The findings obtained in this study and the coping skills prediction system will contribute to the reduction in accidents caused by braking operation mistakes by elderly drivers.

## 7. Conclusions

In this study, the elderly drivers’ reactions were compared to their driving behavior on the simulator to analyze the effects of cognitive and action errors on the braking operation and collision risks. Driving behavior on the simulator was compared with driving behavior at intersections on public roads, and the relationship between driving behavior on public roads and coping skills was investigated. We expanded the scope of the coping skills prediction system and enhanced its usefulness by predicting coping skills from elderly drivers’ preparation for a braking operation near intersections. By preparing for braking operations, the elderly drivers were less likely to have an accident when exposed to the risk of collision in situations with a mental workload. The number of accidents caused by elderly drivers who took behavioral precautions in addition to cognitive precautions was low. The coping skills prediction system predicted coping skills with an accuracy of 70% from elderly drivers’ preparation for a braking operation near intersections. We focused on the left foot posture as a behavioral precaution and provided insights into braking mistakes. The findings obtained in this study and the coping skills prediction system will contribute to the reduction in accidents caused by braking operation mistakes made by elderly drivers.

## Figures and Tables

**Figure 1 sensors-22-05741-f001:**
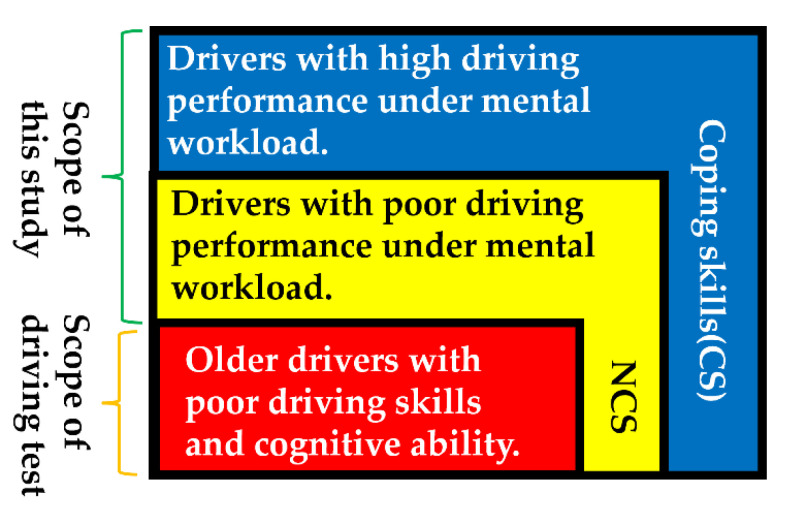
Definition of elderly drivers with coping skills (CS) and without coping skills (NCS) [8].

**Figure 2 sensors-22-05741-f002:**
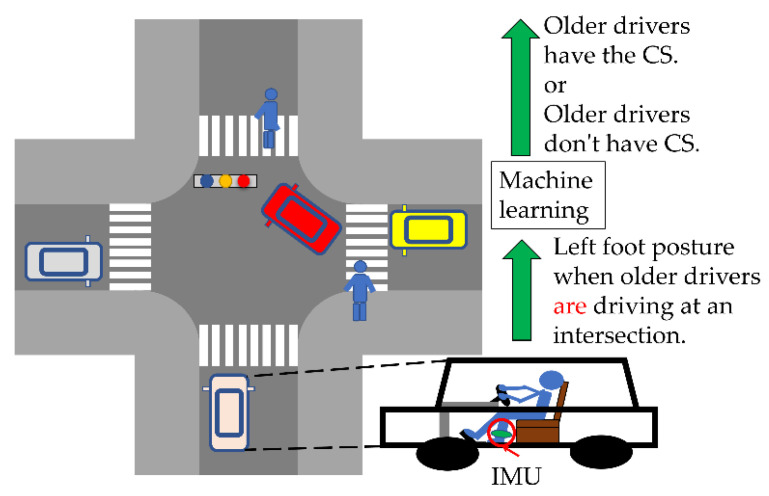
The coping skills prediction (CP) system, in which machine learning predicts coping skills from the left foot movements when an elderly driver drives at an intersection on a public road.

**Figure 3 sensors-22-05741-f003:**
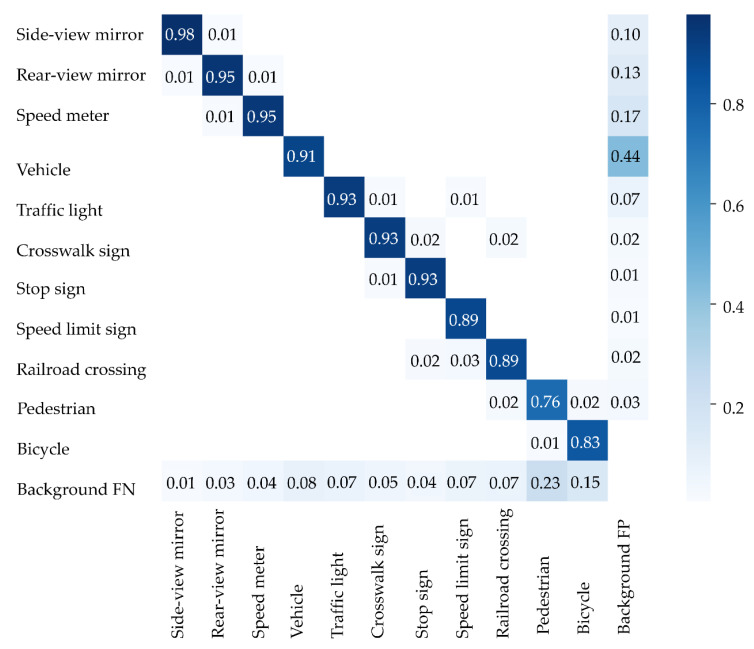
The result of object detection is shown by a confusion matrix.

**Figure 4 sensors-22-05741-f004:**
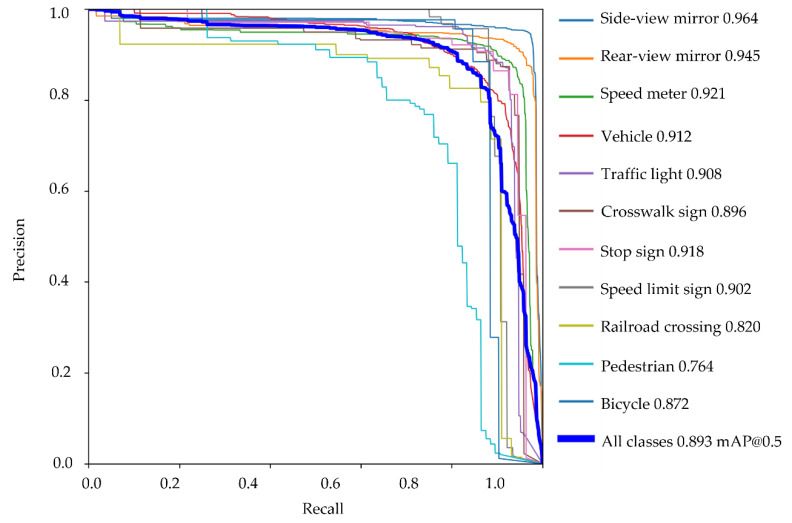
The result of object detection is shown by a PR curve(mAP@0.5).

**Figure 5 sensors-22-05741-f005:**
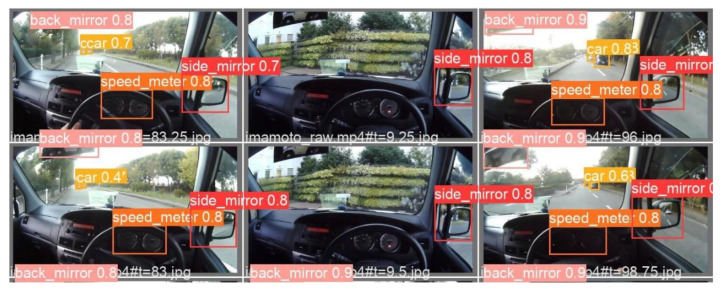
Example of object detection.

**Figure 6 sensors-22-05741-f006:**
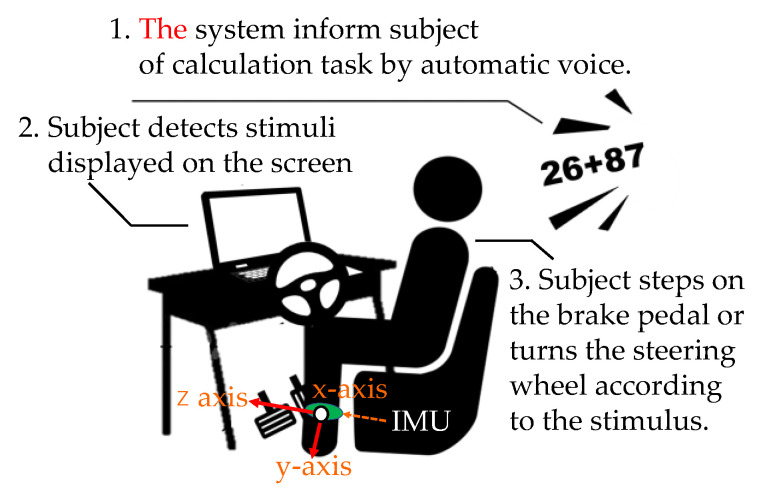
Subject performs dual task.

**Figure 7 sensors-22-05741-f007:**
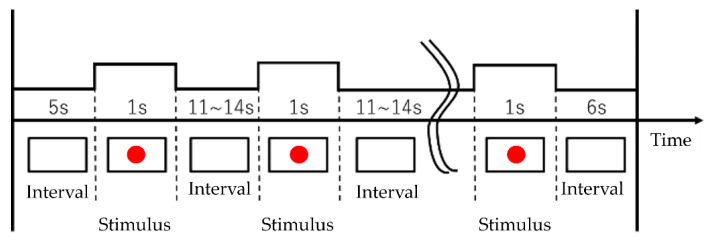
Stimulus interval and presentation.

**Figure 8 sensors-22-05741-f008:**
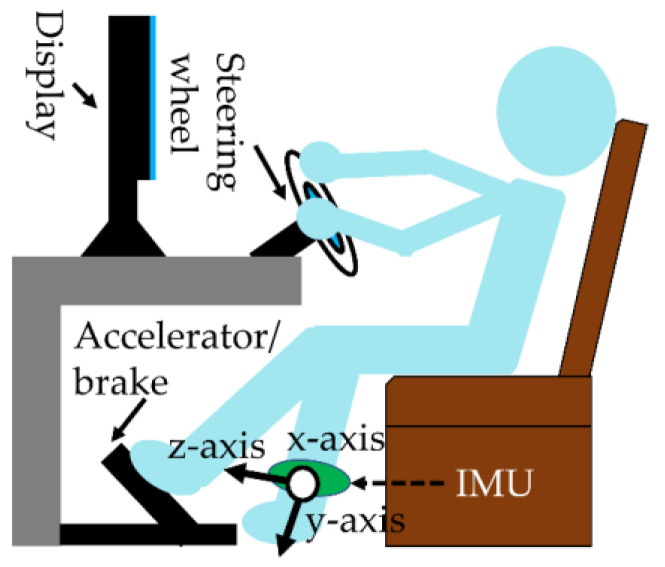
Subject drives on the route with a driving simulator.

**Figure 9 sensors-22-05741-f009:**
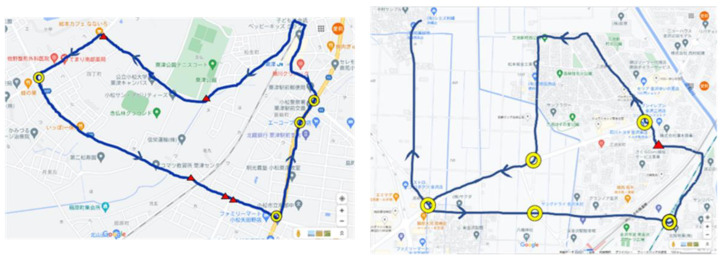
The subject drove route A (**left**) and route B (**right**). The yellow circle on the map means an intersection.

**Figure 10 sensors-22-05741-f010:**
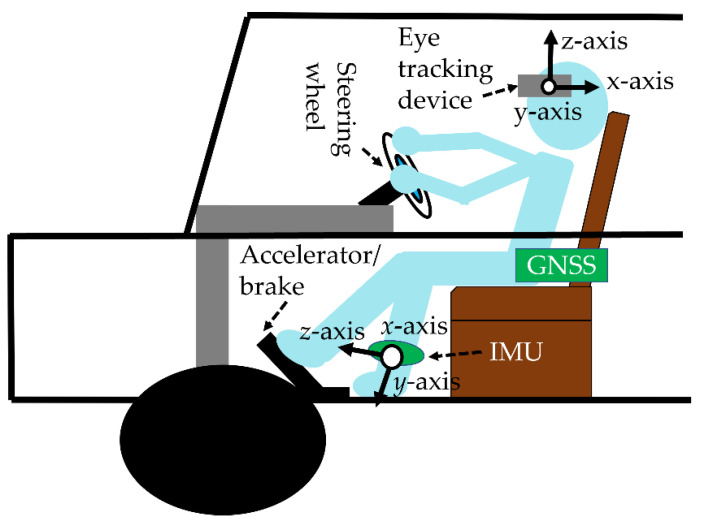
The subjects drive on a public road.

**Figure 11 sensors-22-05741-f011:**
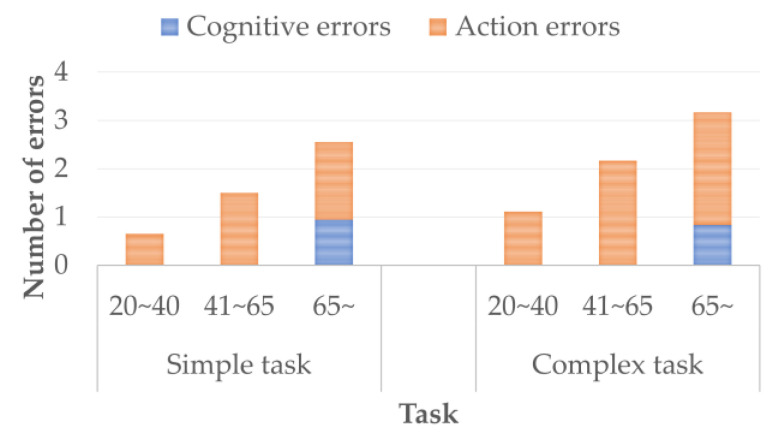
The number of dual task performances is shown by age group.

**Table 1 sensors-22-05741-t001:** The specifications of the measuring equipment.

Device	Model Number	Manufacturer	Measurement	Characteristic Elements	Sampling Frequency
IMU	TSND151	ATR-Promotions	Acceleration, angular velocity, and tilt angle of each axis (*xyz* axis)	Mean, standard deviation (SD), maximum value, and minimum value	50 Hz
Eye tracker	Pupil core	Pupil labs	2D gaze-position when the object appears in the view image		eye camera: 200 Hz, front camera: 30 Hz
GNSS	Smart phone	route history	Vehicle speed		1 Hz
Driving simulator	ACM300	Hitachi KE Systems	The amount of operation of the braking, accelerator, and steering		50 Hz

**Table 2 sensors-22-05741-t002:** Cases where collision accidents are included in the driving route and likely to occur.

Identifier	Case
1	Overtake parked vehicles in the direction of travel.
2	A child jumps out in front of the vehicle.
3	A motorcycle makes a sudden right turn.
4	Sudden braking of the left-turn vehicle.
5	Change the lane of the vehicle in the direction of travel.
6	Parked vehicle after turning left.
7	Sudden braking of vehicles ahead due to the traffic light switching to red.
8	A child jumps out between parked vehicles.
9	Pass by the side of a vehicle parked in the opposite lane.
10	A motorcycle coming straight from the blind spot of a right-turning vehicle in the opposite lane.
11	A taxi in front suddenly brakes to pick up passengers.
12	Older people crossing a road without traffic lights.
13	The rushing of crossing pedestrians when the traffic signal changes.
14	Pedestrians rush when the traffic signal changes.
15	A traffic accident was caused by a driver thanking another driver for letting him go first at a junction.

**Table 3 sensors-22-05741-t003:** The left foot characteristics of few errors and many errors groups in action errors when older drivers perform dual task.

Characteristics	Many Errors		Few Errors		*p*-Value
	Mean	Sd	Mean	Sd	
Mean tilt angle of the left foot on the *y*-axis	51.28	69	−54.91	63.5	0.009

**Table 4 sensors-22-05741-t004:** Dual tasks performance of NCS and CS groups.

	NCS		CS		*p*-Value
	Mean	Sd	Mean	Sd	
Cognitive errors [times]	2.8	3.3	2.2	0.6	0.35
Action errors [times]	2	5.1	1.7	2.3	0.73
Sum of errors [times]	4.8	8.2	3.8	4.2	0.45

**Table 5 sensors-22-05741-t005:** Characteristics of few error and many error groups when the subject drove on the route with a driving simulator.

Type	Case	Characteristics	Many Errors		Few Errors		*p*-Value
			Mean	Sd	Mean	Sd	
Action errors	ALL	Mean of the accelerator operation amount [%]	15.3	7.76	16.3	8.72	0.4914
Action errors	ALL	Maximum of the braking operation amount [%]	63.2	42.4	83	30.8	0.0002
Sum of errors	2	Mean of the accelerator operation amount [%]	14.1	2.56	18.5	2.38	0.0047
Sum of errors	9	Minimum of the steering operation amount [degree]	−482	71.6	−287	129	0.0037

**Table 6 sensors-22-05741-t006:** The amount of operation of the adult driver and elderly driver groups when the subject drove on the route with a driving simulator.

	Elderly Driver		Adult Driver		*p*-Value
	Mean	Sd	Mean	Sd	
Mean of the accelerator operation amount [%]	15.8	8.2	19.2	8.1	0.0000002
SD of the accelerator operation amount [%]	19.4	6.4	23.4	8.2	0.0000008
Mean of the braking operation amount [%]	5.5	6.4	8.6	9	0.0000013
SD of the braking operation amount [%]	14.7	11	19.1	10.2	0.0000691

**Table 7 sensors-22-05741-t007:** Vehicle speed characteristics of elderly drivers and adult drivers when the subject drove on a public road.

	Elderly Driver		Adult Driver		*p*-Value
	Mean	Sd	Mean	Sd	
Mean of the speed [km/h]	35.6	3.5	39.1	4.4	0.02
SD of the speed [km/h]	11.5	3.8	13.8	5.1	0.01

**Table 8 sensors-22-05741-t008:** Eye-gaze and head characteristics of NCS and CS groups when the subject drove on a public road.

	NCS		CS		*p*-Value
	Mean	Sd	Mean	Sd	
The maximum tilt angle of the head on the *z*-axis [degree]	−112	85.6	−173	28.3	0.000007
The minimum tilt angle of the head on the *z*-axis [degree]	140	59.9	178	5.91	0.001744
SD eye gaze on the *x*-axis near the traffic sign	0.23	0.11	0.59	0.03	0.000098
SD eye gaze on the *x*-axis near the moving obstacle	0.23	0.09	0.58	0.03	0.004396

**Table 9 sensors-22-05741-t009:** Left foot characteristics of NCS and CS groups when the subject drove on the intersection on a public road.

	NCS		CS		*p*-Value
	Mean	Sd	Mean	Sd	
Mean tilt angle of the left foot on the *y*-axis [degree]	22.6	78.1	−52.3	84.7	0.00007
Mean tilt angle of the left foot on the *x*-axis [degree]	−57.4	64.2	−86.2	32.3	0.00924
SD tilt angle of the left foot on the *x*-axis [degree]	0.79	0.44	1.22	0.39	0.00002

**Table 10 sensors-22-05741-t010:** Results of identifying CS and NCS by inputting the characteristics of the left foot for machine learning.

	Recall	Precision	F1
Naive Bayes classifier	0.70	0.69	0.69
Random forest	0.71	0.70	0.71

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
