# Peer review of "Effect of Behavioral Precaution on Braking Operation of Elderly Drivers under Cognitive Workloads"

_sensors, 2022, doi:10.3390/s22155741_

Round 1

Reviewer 1 Report

Despite the seemingly extensive research, the results and conclusions are predictable and do not bring significant knowledge to the research area.
Additionally, it is uncertain whether all the data have been properly prepared for analysis and inference (e.g. foot accelerations measured in the vehicle motion).

The analysis of the results is not presented clearly enough in all cases.

Detailed comments:

Lines:

16: (the same sentence in 439): The coping skills prediction system predicted coping skills... -> could you rewrite this sentence without an obvious repetition?

30: We analyzed the driving behavior of an 30 elderly driver on the driving simulator and suggested that the elderly drivers who felt 31 impatience and nervousness could not step to the brake pedal properly [8]. -> In previous work we analyzed ....

32: The number 32 of accidents by elderly drivers was low when they took the behavioral precautions to step 33 on the brakes for an unexpected situation. -> how do you know that?

38: As a result of the experiment, we predicted coping skills with an accuracy of 92% 38 from the posture of the left foot of an elderly driver who felt impatience and nervousness 39 during driving in the simulator. -> This sentence is more appropriate for the summary than for the introduction.

45: When the coping skills prediction system can predict coping skills from driving behavior on 46 public roads, the family will be able to understand changes in the daily coping skills of 47 elderly drivers. -> Why do you write about the 'family'?

52: We analyze the DRT -> please define this abbreviation before first use.

34: We focused on the posture of the left foot as a behavioral precaution., 81: We focus on the left foot posture as a behavioral precaution and provide insights on braking mistakes. This is a repetition.

87: The elderly drivers slow down the reaction time of their driving operations because of the use of mobile phones -> I think that this is general and applies to all drivers regardless of their age.

103: Related work [8] define... -> 'defines' or better 'The authors of work [8] define...'

110: You define abbreviation NCS, but in Fig. 1 you write 'No CS'.

111: As a result of the experiment, we predicted coping skills with an accuracy of 92% -> repetition of phrase from line 38.

Figure 2: 'when older drivers are (not is) driving...'

The phrase 'In this study' appears 9 times in the work and in many places it is unnecessary.

153: The experiment was conducted between October 2022 and December 2022 -> but you submit the paper in June 2022, how it is possible?

154: The experimental design and subjects of this study is completely different from the experimental design and subjects of the related study [8]. In my opinion it is not completely different. For example, you used the same IMU, measured left foot with it, used the same simulator and measured ederly drivers. It is really completely different? You even wrote about it in line 237 and 242: The driving simulator is the same product as the related study [8].

157,158,159: The subjects were 32 drivers the 20 to 64 ages -> ... drivers aged 20 to 64

169: A tilt angle of positive angle on the 169 y-axis means that the drivers turn their toes inwards. A tilt angle of negative angle on the 170 y-axis means that the drivers turn their toes outwards. -> Remove one of these two sentences. The second one follows from the previous one and is obvious.

Table 1: For eye tracker you set sampling frequency 200Hz/30Hz. Why are there two values?

174: We analyze the driver's 2D gaze position when object detection detects -> What 'object detection'? Do you have any software to prepare tis task? Please add some details.

175: We define vehicles, pedestrians, and bicycles as moving obstacles. We define traffic lights, crosswalk signs, stop signs, speed limit signs, and railroad 176 crossings as traffic signs.

What do you mean by "we define'?

178: The product name of the 178 eye tracker is the pupil core made by pupil labs. -> You repeat info for Table 2. Unnecessary.

180: 'and one world camera' - Maybe 'front camera' could be better description.

189: Figure A1-Figure A3 ?

Equation (1) Since the determination of the position has a variable error depending on many conditions, the thus calculated velocity may be inaccurate. Did you use any filtering algorithms?

200: DRT is again defined (it was already defined in line 145)

212: The calculation task is the addition of two or three digits. -> But in Fig. 3 you wrote: 26+87: there are two numbers, but 4 digits.

237/242: In this experiment, the same driving simulator as in the related research [8] was used. -> a repetition.

344: Error! Reference source not found. shows the correlation coefficient -> to be corrected

Title of Table 2.The specifications of the measuring equipment is not clear according the data in thos table.

Table 4: Phrase: Minimum of the steering operation amount, and meaning od column 'Case' is not clear.

Table 6: std instead of sd (as in the previous tables). Do you present in this table mean of the means of the speed? And then mean of the SD of the speed? +/- std? How do you interpret that?

Construction of table 9 is not clear and not adequate to the description in the text.

414: Ohn-Bar et.al [12] estimate attention from the head and the eye-gaze movements and 414 assess collision risk. In other words, the risk of collision is evaluated from the driver's 415 cognitive precaution for unexpected situations. -> You connect this two sentences by 'In other words'. This doesn't match to the case. Similar case is in lines 418-421.

424: We focus on the left foot posture as a behavioral precaution and provide insights on braking mistakes. -> You write almost the same again and again.

Discussion:

Many conclusions are obvious and could be stated without your experiments, e.g. elderly drivers have more cognitive errors than adult drivers, elderly drivers are more likely 361 to miss the traffic signs, it is suggested 390 that elderly drivers adjusted vehicle speed in response to cognitive decline when elderly 391 drivers drove on the public road.

In the simulator, the measured leg accelerations result from its position and the movement of the leg itself. On the other hand, when measuring in motion, accelerations resulting from the motion of the vehicle are added. Has it somehow been taken into account or filtered out?

Therefore, the presented differences in correlations between the simulator and the real car may differ.

Many thoughts, expressions, statements are repeated many times in the work, so reading the article can sometimes irritate the reader. This should be corrected by reviewing the text and removing unnecessary duplications.

The paper is partly similar to [8] Kajiwara, Y., & Kimura, H. (2021). Predicting the coping skills of elderly drivers in the face of unexpected situation. Sensors, 481 21(6), 2099.

You should more clearly explain the relation between the current and the previous paper.

Author Response

Thank you for your review. The paper has been revised according to the review.

Despite the seemingly extensive research, the results and conclusions are predictable and do not bring significant knowledge to the research area.

Additionally, it is uncertain whether all the data have been properly prepared for analysis and inference (e.g. foot accelerations measured in the vehicle motion).

The analysis of the results is not presented clearly enough in all cases.

Detailed comments:

Lines:

16: (the same sentence in 439): The coping skills prediction system predicted coping skills... -> could you rewrite this sentence without an obvious repetition?

103: Related work [8] define... -> 'defines' or better 'The authors of work [8] define...'

110: You define abbreviation NCS, but in Fig. 1 you write 'No CS'.

111: As a result of the experiment, we predicted coping skills with an accuracy of 92% -> repetition of phrase from line 38.

Figure 2: 'when older drivers are (not is) driving...'

The phrase 'In this study' appears 9 times in the work and in many places it is unnecessary.

157,158,159: The subjects were 32 drivers the 20 to 64 ages -> ... drivers aged 20 to 64

169: A tilt angle of positive angle on the 169 y-axis means that the drivers turn their toes inwards. A tilt angle of negative angle on the 170 y-axis means that the drivers turn their toes outwards. -> Remove one of these two sentences. The second one follows from the previous one and is obvious.

175: We define vehicles, pedestrians, and bicycles as moving obstacles. We define traffic lights, crosswalk signs, stop signs, speed limit signs, and railroad 176 crossings as traffic signs.

What do you mean by "we define'?

178: The product name of the 178 eye tracker is the pupil core made by pupil labs. -> You repeat info for Table 2. Unnecessary.

180: 'and one world camera' - Maybe 'front camera' could be better description.

200: DRT is again defined (it was already defined in line 145)

212: The calculation task is the addition of two or three digits. -> But in Fig. 3 you wrote: 26+87: there are two numbers, but 4 digits.

237/242: In this experiment, the same driving simulator as in the related research [8] was used. -> a repetition.

344: Error! Reference source not found. shows the correlation coefficient -> to be corrected

Title of Table 2.The specifications of the measuring equipment is not clear according the data in thos table.

424: We focus on the left foot posture as a behavioral precaution and provide insights on braking mistakes. -> You write almost the same again and again.

Answer:

I reviewed the expression and corrected the text.

30: We analyzed the driving behavior of an elderly driver on the driving simulator and suggested that the elderly drivers who felt impatience and nervousness could not step to the brake pedal properly [8]. -> In previous work we analyzed ....

Answer:

It was a misunderstanding, so I corrected it.

38: As a result of the experiment, we predicted coping skills with an accuracy of 92% from the posture of the left foot of an elderly driver who felt impatience and nervousness during driving in the simulator. -> This sentence is more appropriate for the summary than for the introduction.

Answer:

This text explains the results of past studies [8]. It was a misunderstanding, so I corrected it.

45: When the coping skills prediction system can predict coping skills from driving behavior on public roads, the family will be able to understand changes in the daily coping skills of elderly drivers. -> Why do you write about the 'family'?

Answer:

I have posted it as an example of using the system. However, it was a misunderstanding, so I deleted it.

52: We analyze the DRT -> please define this abbreviation before first use.

Answer:

Dual task was a more accurate word than DRT. Changed DRT to a dual task word.

34: We focused on the posture of the left foot as a behavioral precaution., 81: We focus on the left foot posture as a behavioral precaution and provide insights on braking mistakes. This is a repetition.

Removed duplicate text.

87: The elderly drivers slow down the reaction time of their driving operations because of the use of mobile phones -> I think that this is general and applies to all drivers regardless of their age.

Answer:

Changed Elderly driver to driver.

153: The experiment was conducted between October 2022 and December 2022 -> but you submit the paper in June 2022, how it is possible?

Answer:

It was a typo. The experiment was conducted between October 2021 and December 2021.

154: The experimental design and subjects of this study is completely different from the experimental design and subjects of the related study [8]. In my opinion it is not completely different. For example, you used the same IMU, measured left foot with it, used the same simulator and measured ederly drivers. It is really completely different? You even wrote about it in line 237 and 242: The driving simulator is the same product as the related study [8].

Answer:

It was a misunderstanding.

The subjects of this study is completely different from subjects of the related study [8]. Regarding the experimental design, the acquisition of driving behavior on the simulator was performed in the same procedure as in the related research [8]. Experimental design of driving behavior in dual tasks and driving behavior in the real world is an experiment added and carried out in this study.

Table 1: For eye tracker you set sampling frequency 200Hz/30Hz. Why are there two values?

Answer:

200Hz is the sampling rate of the eye camera and 30Hz is the sampling rate of the front camera.

174: We analyze the driver's 2D gaze position when object detection detects -> What 'object detection'? Do you have any software to prepare tis task? Please add some details.

Answer:

We detect the objects in the view image using machine learning. The object detection algorithm is yolov5. The yolov5 program was downloaded from git hub (https://github.com/ultralytics/yolov5). The program was executed using Visual studio code. The Python version is 3.7. The computer was equipped with GPU: RTX3080.

189: Figure A1-Figure A3 ?

Figures.A1 to Figure.A3 have been modified to Figure 3 to Figure 5.

Equation (1) Since the determination of the position has a variable error depending on many conditions, the thus calculated velocity may be inaccurate. Did you use any filtering algorithms?

In order to reduce noise, a moving average filter was applied to the time series data calculated by Eq. (1). The time window is 3 seconds.

Table 4: Phrase: Minimum of the steering operation amount, and meaning od column 'Case' is not clear.

An accident is a collision with a pedestrian or a vehicle. The amount of steering operation is measured from -360 degrees to 360 degrees. In addition, the brake and accelerator represent the depressed state from 0% to 100%. Added Table.2.

Table 6: std instead of sd (as in the previous tables). Do you present in this table mean of the means of the speed? And then mean of the SD of the speed? +/- std? How do you interpret that?

Construction of table 9 is not clear and not adequate to the description in the text.

Answer:

SD is the standard deviation. I deleted Table 9 and re-explained it in the text. The correlation coefficient of the average tilt angle of the left foot on the y-axis between driving in the simulator and driving in the real world was a moderate correla-tion (= 0.44). The correlation coefficient of the average tilt angle of the left foot on the x-axis between driving in the simulator and driving in the real world was a moderate correlation (= 0.51). The correlation coefficient of the average tilt angle of the left foot on the left y-axis between driving in the simulator and the dual task was a strong cor-relation (= 0.84). The correlation coefficient of the average tilt angle of the left foot on the x-axis between driving in the simulator and the dual task was a strong correlation (= 0.99).

414: Ohn-Bar et.al [12] estimate attention from the head and the eye-gaze movements and 414 assess collision risk. In other words, the risk of collision is evaluated from the driver's 415 cognitive precaution for unexpected situations. -> You connect this two sentences by 'In other words'. This doesn't match to the case. Similar case is in lines 418-421.

Answer:

It was a misunderstanding, so I corrected it.

Discussion:

Many conclusions are obvious and could be stated without your experiments, e.g. elderly drivers have more cognitive errors than adult drivers, elderly drivers are more likely 361 to miss the traffic signs, it is suggested 390 that elderly drivers adjusted vehicle speed in response to cognitive decline when elderly 391 drivers drove on the public road.

In the simulator, the measured leg accelerations result from its position and the movement of the leg itself. On the other hand, when measuring in motion, accelerations resulting from the motion of the vehicle are added. Has it somehow been taken into account or filtered out?

Therefore, the presented differences in correlations between the simulator and the real car may differ.

Many thoughts, expressions, statements are repeated many times in the work, so reading the article can sometimes irritate the reader. This should be corrected by reviewing the text and removing unnecessary duplications.

The paper is partly similar to [8] Kajiwara, Y., & Kimura, H. (2021). Predicting the coping skills of elderly drivers in the face of unexpected situation. Sensors, 481 21(6), 2099.

You should more clearly explain the relation between the current and the previous paper.

Answer:

Section 2.2 was added to clarify the difference from the previous study [8].

The contributions of this study are as follows.

  • We clarify the effect of behavioral precaution on braking operation of elderly drivers during cognitive workloads.
  • We provide new insights into behavioral precautions for older drivers' braking operations in the real world.
  • We predicted coping skills from natural driving behavior near intersections in the real world.

Related study [9] reported that the driver's braking operation slows down when a driver drives with a cognitive workload. However, it has not been validated for behav-ioral precaution when braking is impaired by cognitive workloads. This study analyzes the dual task performance of older drivers and reveals the effects of behavioral pre-vention on driver braking under cognitive workloads.

It is difficult to perfectly reproduce the real-world situation on the simulator. For example, when the driver accelerates the car in the real world, the driver receives the acceleration and stabilizes the body with his left foot. In addition, the position and posture of the left foot of the elderly driver will differ depending on the vehicle type. This study compares driving behavior on a simulator with driving behavior in the real world. It provides insights into real-world behavioral precautions not verified in related studies [8].

Elderly drivers are most likely to mistake the brake and accelerator when starting from a stopped state [1]. When a driver drives near an intersection, he / she feels cog-nitive stress, impatience, and tension. For this reason, we are focusing on preparing for braking near intersections. In related study [8], coping skills could not be predicted without driving in a special environment such as a simulator. We expanded the range of use of the coping skill prediction system and enhanced its usefulness by making it possible to predict from the natural driving behavior near the intersection.

We consider the predictability of coping skills from driving behavior at intersec-tions on public roads. From Table 10, the correlation coefficient of the left foot posture between the driving on the simulator and the driving on the public road was a medium correlation. In addition, the correlation coefficient of the left foot posture between the driving on the simulator and the dual tasks was a strong correlation. The reason why the posture of the left foot when driving in the simulator and driving in the real world had a medium correlation is considered to be the difference in the environment inside the vehicle. The posture of the left foot is calculated from the accelerometer, angular veloc-ity, and geomagnetic sensor. Therefore, it is possible that the acceleration of the car was mixed in with the value acquired by the accelerometer, causing an error in the calcula-tion of the posture of the left foot, but this effect was very small because the vehicle was driven at low speed near the intersection. From Table 9, CS in elderly drivers turned their left foot toes outward. To adapt to the cognitive decline, the elderly drivers with a small sum of errors in dual tasks turned their left foot toes outward to create space for the right foot to step on the brake pedal. It can be expected to elderly drivers are less likely to have an accident on the public road when they are exposed to the risk of colli-sion in situations with a mental workload by taking behavioral precautions. In addition, the mean tilt angle of the X-axis of the left foot is perpendicular to the ground in CS than in NCS. On the other hand, the SD of the X-axis tilt angle of the left foot is larger in CS than in NCS, and the body is unstable. However, it was not possible to clarify whether this difference in the inclination angle of the left foot directly causes a collision in the real world.

Reviewer 2 Report

This paper deals with observation of driver behavior in different experimental environments with the goal of detecting appropriate precaution actions or coping skills. The main problem of the manuscript is a very poor level of English language and the text is hard to understand. Several sentences are repeated several times making it very difficult to follow some concepts described in the paper. Particularly the text in the Results and Discussion section needs significant improvements. There are also lots of typos and grammatical errors. I would strongly recommend substantial revision of the English language to be able to do more comprehensive review of the described work.

The content of this paper is closely related to topic described in another paper by the same author: Kajiwara, Y., & Kimura, H. (2021). Predicting the coping skills of elderly drivers in the face of unexpected situation. Sensors, 481 21(6), 2099. It is not clear how the new paper extends the “Coping Skills Prediction System” from the original paper. Figures 1 and 2 are very similar in both papers… I suggest adding a separated paragraph on explicitly explaining the new contributions of this work compared to the previous paper.

Additional comments:

1.       Lines 144-148: The experiments described in this paper include DRT, simulated driving and driving on public roads. How where these three different experiments selected, what is the rationale behind it?

2.       Line 153: October and December 2022??

3.       Line 203: Authors describe their “cognitive workload” experiment as DRT which is not correct. DRT (Detection-Reponse Task) is a standardized method for observing the effect of cognitive load described in “ISO (International Organization for Standardization). Road Vehicles—Transport Information and Control Systems—Detection-Response Task (DRT) for Assessing Attentional Effects of Cognitive Load in Driving; ISO: Geneva, Switzerland, 2016; p. 17488”. Authors should make clear distinction between the modified method that they are using and the original standard.

4.       Line 214: Was the result of this cognitive task (calculating the formula) evaluated at all? Did test subjects have to report it to the experimenter?

5.       Line 241: What is the threshold for this classification? Was it done manually or with an algorithm?

6.       Line 246: A more detailed description of the simulated driving is needed. Where is Table 2 with these 15 cases?

7.       Line 248. A definition of an accident should be provided. What is considered as an accident?

8.       Line 258: How was this “5 km/h” threshold defined? Why not really stopped?

9.       Line 265: Typically, the main reason for using non-parametric test such as Mann-Whitney U Test is non-normal distribution of the dependent variable? Was this the case here? How was the (non)normality checked?

10.   Line 270: The rejection region for null hypothesis set by authors is 0.01. Later in the text they compare their p level to 0.05. Which is the correct significance level?

11.   Line 276: What is the reason for this specific threshold?

12.   Line 322: Sections 5.2 and 5.3 have the same title.

13.   Methodology: It is not clear which dependent variables were measured in each of the three experiments. I suggest adding a table of variables/measures for each section (DRT, simulator, on-road driving) by providing also some background what individual variable represents (what can be observed from it).

14.   Line 313: Table 4 presents results from the simulator but it is not clear what are these characteristics (e.g. mean of the accelerator operation amount, maximum of the braking operation amount, etc). Some further explanations should be provided.

15.   Results: All sections of the results should also provide some interpretations. It is difficulty to understand the results in the table without any explanations at all. Also, units of values in the tables should be provided (e.g. what is the unit for “SD eye gaze on the x-axis near the traffic sign”?). What does this parameter tell us?

Author Response

Thank you for your review. The paper has been revised according to the review.

This paper deals with observation of driver behavior in different experimental environments with the goal of detecting appropriate precaution actions or coping skills. The main problem of the manuscript is a very poor level of English language and the text is hard to understand. Several sentences are repeated several times making it very difficult to follow some concepts described in the paper. Particularly the text in the Results and Discussion section needs significant improvements. There are also lots of typos and grammatical errors. I would strongly recommend substantial revision of the English language to be able to do more comprehensive review of the described work.

The content of this paper is closely related to topic described in another paper by the same author: Kajiwara, Y., & Kimura, H. (2021). Predicting the coping skills of elderly drivers in the face of unexpected situation. Sensors, 481 21(6), 2099. It is not clear how the new paper extends the “Coping Skills Prediction System” from the original paper. Figures 1 and 2 are very similar in both papers… I suggest adding a separated paragraph on explicitly explaining the new contributions of this work compared to the previous paper.    

Answer:

We used the English correction service and drastically revised the English expressions.

Section 2.2 was added to clarify the difference from the previous study [8].

The contributions of this study are as follows.

  • We clarify the effect of behavioral precaution on braking operation of elderly drivers during cognitive workloads.
  • We provide new insights into behavioral precautions for older drivers' braking operations in the real world.
  • We predicted coping skills from natural driving behavior near intersections in the real world.

Related study [9] reported that the driver's braking operation slows down when a driver drives with a cognitive workload. However, it has not been validated for behav-ioral precaution when braking is impaired by cognitive workloads. This study analyzes the dual task performance of older drivers and reveals the effects of behavioral pre-vention on driver braking under cognitive workloads.

It is difficult to perfectly reproduce the real-world situation on the simulator. For example, when the driver accelerates the car in the real world, the driver receives the acceleration and stabilizes the body with his left foot. In addition, the position and posture of the left foot of the elderly driver will differ depending on the vehicle type. This study compares driving behavior on a simulator with driving behavior in the real world. It provides insights into real-world behavioral precautions not verified in related studies [8].

Elderly drivers are most likely to mistake the brake and accelerator when starting from a stopped state [1]. When a driver drives near an intersection, he / she feels cog-nitive stress, impatience, and tension. For this reason, we are focusing on preparing for braking near intersections. In related study [8], coping skills could not be predicted without driving in a special environment such as a simulator. We expanded the range of use of the coping skill prediction system and enhanced its usefulness by making it possible to predict from the natural driving behavior near the intersection.

Additional comments:

  1. Lines 144-148: The experiments described in this paper include DRT, simulated driving and driving on public roads. How where these three different experiments selected, what is the rationale behind it?

Answer:

Theoretical background of Dual task

The subject solves the calculation task and at the same time performs the driving behavior according to the visual stimulus shown on the display. Calculation tasks are processed in working memory [20] and long-term memory. Working memory consists of a central executive and a slave system. The central execution controls attention, in-tegrates information, and manages slave systems. The slave system temporarily retains and manipulates information. The slave system consists of a phonological loop, visuospatial sketchpad, and episode buffer. As the driver ages, the associative memory, memory retention / retrieval, attention suppression function, and attention target up-date decline [21]-[23]. The performance of the dual task represents an overall evaluation of these functions.

Simulator

Regarding the experimental design, the acquisition of driving behavior on the simulator was performed in the same procedure as in the related research [8]. Experimental design of driving behavior in dual tasks and driving behavior in the real world is an experiment added and carried out in this study. Elderly drivers were classified into CS and NCS groups based on the number of accidents while driving in a driving simulator, using the same thresholds as in the re-lated study [8]. In the related study, the average number of accidents on the simulator was 1.6 ± 1.1, so subjects with 2 or less accidents were divided into CS and other subjects were divided into NCS. In this experiment, CS and NCS are clasified using the same threshold value (= 2) as in the related studies.

Driving on public roads

It is difficult to perfectly reproduce the real-world situation on the simulator. For example, when the driver accelerates the car in the real world, the driver receives the acceleration and stabilizes the body with his left foot. In addition, the position and posture of the left foot of the elderly driver will differ depending on the vehicle type. This study compares driving behavior on a simulator with driving behavior in the real world. It provides insights into real-world behavioral precautions not verified in related studies [8].

  1. Line 153: October and December 2022??

Answer:

It was a typo. The experiment was conducted between October 2021 and December 2021.

  1. Line 203: Authors describe their “cognitive workload” experiment as DRT which is not correct. DRT (Detection-Reponse Task) is a standardized method for observing the effect of cognitive load described in “ISO (International Organization for Standardization). Road Vehicles—Transport Information and Control Systems—Detection-Response Task (DRT) for Assessing Attentional Effects of Cognitive Load in Driving; ISO: Geneva, Switzerland, 2016; p. 17488”. Authors should make clear distinction between the modified method that they are using and the original standard.

Answer:

Dual task was a more accurate word than DRT. Changed DRT to a dual task word.

  1. Line 214: Was the result of this cognitive task (calculating the formula) evaluated at all? Did test subjects have to report it to the experimenter?

Answer:

The subject is cognitively burdened when a subject solves a calculation task. Subjects calculated the addition of two or three digits. The subject listens to the formula played by the voice recorder and calculates it in his head according to the formula. If the subject was able to calculate in time, the subject verbally responded with the calculation result. If the subject was unable to calculate in time, he / she was instructed to preferentially solve the next calculation task.

  1. Line 241: What is the threshold for this classification? Was it done manually or with an algorithm?
  2. Line 276: What is the reason for this specific threshold?

Answer:

Subjects in the elderly driver group are divided into the NCS group and CS group. In the related study, the average number of accidents on the simulator was 1.6 ± 1.1, so subjects with 2 or less accidents were divided into CS and other subjects were divided into NCS. In this experiment, CS and NCS are clasified using the same threshold value (= 2) as in the related studies.

  1. Line 246: A more detailed description of the simulated driving is needed. Where is Table 2 with these 15 cases?

Answer:

Added Table.2.

  1. Line 248. A definition of an accident should be provided. What is considered as an accident?

Answer:

An accident is a collision with a pedestrian or a vehicle.

  1. Line 258: How was this “5 km/h” threshold defined? Why not really stopped?

Answer:

On the other hand, since the value measured by GPS contains noise, "vehicle stop" in this study means that the vehicle speed is 5 km / h or less.

  1. Line 265: Typically, the main reason for using non-parametric test such as Mann-Whitney U Test is non-normal distribution of the dependent variable? Was this the case here? How was the (non)normality checked?

Answer:

In this study, we analyze the characteristic of drivers in the Mann-Whitney U Test in the hypothesis test. When performing a test, the multiplicity of tests is often a problem. In this study, we adopt a nonparametric test to avoid the problem of multiplicity.

  1. Line 270: The rejection region for null hypothesis set by authors is 0.01. Later in the text they compare their p level to 0.05. Which is the correct significance level?

Answer:

The rejection region is 0.01. When the null hypothesis is rejected, a variable has a significant difference between the two groups. However, it has been pointed out that the significance level is not an absolute indicator [25]. Therefore, the characteristic elements with p-value less than 0.05 are also analyzed.

  1. Line 322: Sections 5.2 and 5.3 have the same title.

Answer:

I reviewed the expression and corrected the text.

  1. Methodology: It is not clear which dependent variables were measured in each of the three experiments. I suggest adding a table of variables/measures for each section (DRT, simulator, on-road driving) by providing also some background what individual variable represents (what can be observed from it).

Answer:

The features and the names of experiments using each device have been added to Table 1.

  1. Line 313: Table 4 presents results from the simulator but it is not clear what are these characteristics (e.g. mean of the accelerator operation amount, maximum of the braking operation amount, etc). Some further explanations should be provided.

Answer:

An accident is a collision with a pedestrian or a vehicle. The amount of steering operation is measured from -360 degrees to 360 degrees. In addition, the brake and accelerator represent the depressed state from 0% to 100%.

  1. Results: All sections of the results should also provide some interpretations. It is difficulty to understand the results in the table without any explanations at all. Also, units of values in the tables should be provided (e.g. what is the unit for “SD eye gaze on the x-axis near the traffic sign”?). What does this parameter tell us?

Answer:

SD is the standard deviation.

The 2D eye-gaze coordinates are represented in the normalized image coordinate system. The center of the view image is 0 on the XY axes of the normalized image co-ordinate system. The X-axis represents horizontal eye-gaze movement. The Y-axis rep-resents vertical eye-gaze movement. The 2D eye-gaze position is represented in the range -1 to 1 on the XY axes.

Round 2

Reviewer 1 Report

The authors took into account the comments and made appropriate corrections to the text.

Author Response

Thank you for your review.

Reviewer 2 Report

Authors have provided significantly improved manuscript, I really appreciate the professional English proofreading. I believe they have successfully addressed big majority of my comments, however there are two points where I don’t agree with authors I would appreciate some further but minor edits.

I am providing bellow my original comments and my new comments based on authors’ responses:

My original comment:

Line 265: Typically, the main reason for using non-parametric test such as Mann-Whitney U Test is non-normal distribution of the dependent variable? Was this the case here? How was the (non)normality checked?

Authors’ response:

In this study, we analyze the characteristic of drivers in the Mann-Whitney U Test in the hypothesis test. When performing a test, the multiplicity of tests is often a problem. In this study, we adopt a nonparametric test to avoid the problem of multiplicity.

 My new comment:

Mann-Whitney U Test is a statistical test for comparing two samples when variables have not-normal distribution. I don’t see how this is related to “multiplicity” pointed out by the authors. I don’t see any explanation or information on the data distribution here. Typically this is performed with Shapiro-Wilk Normality test (or something similar) and then decided whether to use two sample t-tests (in case of normal distribution) or Mann-Whitney (in case of non-normal distribution).  When more groups of data are in question (I believe this is what authors call multiplicity) one should consider ANOVA or Kruskal-Wallis tests (again depending on data normality). Perhaps I am missing or not understanding something here and I would ask authors to provide clarifications.

My original comment:

Results: All sections of the results should also provide some interpretations. It is difficulty to understand the results in the table without any explanations at all. Also, units of values in the tables should be provided (e.g. what is the unit for “SD eye gaze on the x-axis near the traffic sign”?). What does this parameter tell us?

Authors responses:

SD is the standard deviation.

The 2D eye-gaze coordinates are represented in the normalized image coordinate system. The center of the view image is 0 on the XY axes of the normalized image co-ordinate system. The X-axis represents horizontal eye-gaze movement. The Y-axis rep-resents vertical eye-gaze movement. The 2D eye-gaze position is represented in the range -1 to 1 on the XY axes.

 My new comment:

I am asking about the “unit” for all these variables and not about their meaning. For example, what is the unit in:

-          Table 6: Is this amount in %, time, distance?  

-          Table 7: km/h or m/h?

-          Table 8: “SD eye gaze on the x-axis near the traffic sign”: it seems this is not in mm or pixels but some sort of normalized value in the range from -1 to 1.

-          Table 9: “Mean tilt angle of the left foot on the y-axis, etc: degree, rad, grad?

I suggest adding this information to the captions of the tables in the manuscript.

Author Response

Authors have provided significantly improved manuscript, I really appreciate the professional English proofreading. I believe they have successfully addressed big majority of my comments, however there are two points where I don’t agree with authors I would appreciate some further but minor edits.

I am providing bellow my original comments and my new comments based on authors’ responses:

  1. Mann-Whitney U Test is a statistical test for comparing two samples when variables have not-normal distribution. I don’t see how this is related to “multiplicity” pointed out by the authors. I don’t see any explanation or information on the data distribution here. Typically this is performed with Shapiro-Wilk Normality test (or something similar) and then decided whether to use two sample t-tests (in case of normal distribution) or Mann-Whitney (in case of non-normal distribution). When more groups of data are in question (I believe this is what authors call multiplicity) one should consider ANOVA or Kruskal-Wallis tests (again depending on data normality). Perhaps I am missing or not understanding something here and I would ask authors to provide clarifications.

Answer:

Many researchers test both the normally distributed population and equal SDs assumptions through statistical significance tests (e.g., Fasano & Francheschini, 1987; Justel, Peña, & Zamar, 1997; Levene, 1960; Shapiro & Wilk, 1965; Smirnov, 1948; Stephens, 1974). However, this approach is not without problems.

Statisticians used to recommend testing for it and if the assumption was violated, use an adjustment to correct for it. However, more recently statisticians have stopped using this approach, for two reasons. First, violating this assumption only matters if you have unequal group sizes; if you don’t have unequal group sizes, the assumption is pretty much irrelevant and can be ignored. Second, the tests of equal SDs assumptions and the normally distributed population tend to work very well when you have large samples (when it doesn’t matter as much if you have violated the assumption) smaller samples - which is exactly when it matters.

[A]Leppink, J. (2019). Statistical methods for experimental research in education and psychology. Cham: Springer.

[B]Field, A., Miles, J., & Field, Z. (2012). Discovering statistics using R. Sage publications.

Since the sample size of this experiment is 30 or less, the reliability of the results is low even if the test of normally distributed population and equal SDs assumptions rejected hypothesis. If the data distribution is normally distributed, the power will be reduced if the U test is applied, but there is no problem if the U test is applied to the data of the normal distribution. Also, as mentioned above, even if the normality test hypothesis is rejected, it does not prove to be a normal distribution. It is difficult to specify the distribution of the data. In addition, in general, driving behavior data is not normally distributed. For the above reasons, we used a nonparametric test in this study.

These problems are referred to as the problem of multiplicity in this paper.

  1. I am asking about the “unit” for all these variables and not about their meaning. For example, what is the unit in:

-          Table 6: Is this amount in %, time, distance? 

-          Table 7: km/h or m/h?

-          Table 8: “SD eye gaze on the x-axis near the traffic sign”: it seems this is not in mm or pixels but some sort of normalized value in the range from -1 to 1.

-          Table 9: “Mean tilt angle of the left foot on the y-axis, etc: degree, rad, grad?

I suggest adding this information to the captions of the tables in the manuscript.

Answer:

Added units to Table 5-9.

There is no unit of “SD eye gaze on the x-axis near the traffic sign”.

This manuscript is a resubmission of an earlier submission. The following is a list of the peer review reports and author responses from that submission.